# Phenotypic, Physiological, and Gene Expression Analysis for Nitrogen and Phosphorus Use Efficienies in Three Popular Genotypes of Rice (*Oryza sativa* Indica)

**DOI:** 10.3390/plants13182567

**Published:** 2024-09-13

**Authors:** Bhumika Madan, Nandula Raghuram

**Affiliations:** Centre for Sustainable Nitrogen and Nutrient Management, University School of Biotechnology, Guru Gobind Singh Indraprastha University, Dwarka, New Delhi 110078, India; bhumika2302@gmail.com

**Keywords:** crop nutrient use efficiency, nitrogen, phosphorus, NUE, PUE, rice, partial factor productivity, N-response, P-response, harvest index

## Abstract

Crop nitrogen (N) and phosphorus (P) use efficiencies (NUE/PUE) are important to minimize wastage and nutrient pollution, but no improved crop for both is currently available. We addressed them together in rice, in the view of its high consumption of NPK fertilizers. We analyzed 46 morphophysiological parameters for the N/P response in three popular indica genotypes, namely, BPT 5204, Panvel 1, and CR Dhan 301 at low, medium, and normal N/P doses. They include 18 vegetative, 15 physiological, and 13 reproductive parameters. The segregation of significantly N/P-responsive parameters correlating with NUE/PUE revealed 21 NUE, 22 PUE, and 12 common parameters. Feature selection analyses revealed the common high-ranking parameters including the photosynthetic rate at the reproductive stage, tiller number, root–shoot ratio, culm thickness, and flag leaf width. The venn selection using the reported NUE/PUE-related candidate genes in rice revealed five genes in common for both, namely *OsIAA3*, *OsEXPA10*, *OsCYP75B4*, *OsSultr3;4*, and *OsFER2*, which were associated with three of the common traits for NUE/PUE. Their expression studies using qRT-PCR revealed the opposite regulation in contrasting genotypes for *OsSultr3;4* and *OsEXPA10* in N-response and for *OsFER2* in P-response, indicating their role in contrasting N/P use efficiencies. Overall, CR Dhan 301 has the highest NUE and PUE followed by Panvel 1 and BPT5204 among the studied genotypes.

## 1. Introduction

Nitrogen (N) and phosphorus (P) compounds are major macronutrients often supplied as inorganic fertilizers to boost plant growth and crop yield [1] and are credited to have hugely contributed to global food security. However, yield-centric crop improvement under high-input conditions resulted in low fertilizer use efficiency for most of the crops, at around 30% for N [2,3] and 25% for P [4], while the rest are lost to the environment. N losses occur in the form of NH_3_, N_2_O, and N_2_ lost to the air and as NO_3_^−^ and NH_4_^+^ compounds lost to water [5], while phosphates are only lost to water [6]. They cause air pollution, ill health, climate change, the eutrophication of water bodies, biodiversity loss, etc., increasingly threatening the sustainability of planet earth [7]. The target 7 of the intergovernmental Kunming–Montreal Global Biodiversity Framework (2022) mandates countries to halve nutrient waste by 2030.

Even though N/P use efficiency (NUE/PUE) can be improved to some extent by using better fertilizers or agronomic practices, crop genetic improvement is critical to further enhance NUE/PUE and limit the economic and environmental costs of unused nutrients [8]. Nutrient use efficiency can be measured in many ways, including biomass/output/grain yield per unit input, harvest index, or partial factor productivity, and it is common to use multiple indices of measurement [2,9]. The complete mechanisms for NUE/PUE have not yet been understood despite recent progress [10,11,12,13,14].

Crop improvement efforts towards efficiency emphasize developing genotypes that provide similar or higher yields under lower nutrient doses. Its major requirements include characterization of the phenotypic traits, donor/recipient genotypes, quantitative trait loci (QTL), breeding populations, etc., for forward genetics and genomic/functional genomic resources and mutants and transgenic/genome-edited plants for reverse genetics. Various agronomic, transgenic, and QTL-based approaches are being used to increase the nitrogen use efficiency of crops [9,15,16,17,18,19]. Similarly, efforts to improve PUE in various crops have been reviewed [13,20,21] However, there is currently no field crop in which both NUE and PUE have been improved, even though both efficiencies are often needed in the same crop.

Rice is among the world’s top three most produced crops and is the top-most in India, with concomitantly high consumption of N/P fertilizers. It is also the most attractive target for crop improvement due to its low nutrient use efficiency, huge germplasm, and genomic resources [22]. The phenotype for NUE has been better characterized in rice, and contrasting NUE genotypes have been identified [23,24]. The complete transcriptomes are known in rice for nitrate response in Indica [25] and Japonica [26], apart from the urea response in contrasting Indica genotypes [27]. Hierarchical shortlisting of candidate genes for NUE has been performed on various criteria [22,28]. Efforts to improve NUE in rice have been reviewed recently [9,14,16,29] including the genetic manipulation of transporters and transcription factors [30,31,32,33,34], N-metabolizing enzymes [35], and others involved in autophagy, epigenetic regulation, photosynthesis, stress, and signaling, with varying degrees of success [11,36,37,38]. Similarly, the traits for PUE and efforts to improve PUE in rice have been well-documented [39,40], including the genetic manipulation of some transporters, kinases, phosphatases, phosphodiesterases, transcription factors, and others, even as new candidate genes keep emerging [13,41,42,43,44].

The current study addressed NUE and PUE together for the first time via morphophysiological evaluation of the N/P response and candidate gene expression in three popular indica genotypes of rice, namely, BPT 5204, Panvel 1, and CR Dhan 301. They were chosen based on their high yield, their cultivation in different agroclimatic zones of India, and prior literature (Appendix A). We identified some common phenotypic/physiological traits common to NUE and PUE and validated the differential gene expression of five candidate genes for both.

## 2. Results

### 2.1. Unique and Common N/P-Responsive Parameters

Out of 46 measured parameters spanning vegetative and reproductive phenotypic traits as well as physiological parameters (Appendix A), only 26 parameters listed in Table 1 were significantly N-responsive. They exhibited clear dose-response patterns in all three genotypes, even if they had significant genotype-specific differences (Appendix A). Similarly, out of the 46 measured parameters for the P-response, only 31 parameters (Table 2) were significantly P-responsive with clear dose–response patterns in all three genotypes, even if they had significant genotype-specific differences (Appendix A). There were 19 common parameters showing a significant dose–response pattern for both N and P, including most of the vegetative parameters, number of panicles, 50% flowering time, partial factor productivity, and harvest index or HI (Appendix A). Overall, 37 out of 46 measured phenotypic parameters showed significant differences between genotypes at the same dose (Appendix A).

### 2.2. Yield-Associated N/P-Responsive Parameters

The unique as well as common but significant N/P-responsive parameters were shortlisted based on their correlation with yield. Filled seed weight was used as the yield parameter for this purpose, as well as for the purpose of calculating NUE/PUE. Figure 1 shows the scatter plots of the mean values of each of the parameters from three genotypes at normal, medium, and low doses of N and their correlation with corresponding yield values. Out of the 26 N-responsive parameters, 22 were significantly correlated with yield at a *p*-value of <0.05. In addition, leaf chlorophyll content at the vegetative stage and leaf width at the reproductive stage were correlated with the yield at a less significant level (*p*-value < 0.1) (Appendix A).

Figure 2 shows the scatter plots of the mean values of each of the parameters from three genotypes at normal, medium, and low doses of P and their correlation with corresponding yield values. Out of 31 P-responsive parameters, 17 were significantly correlated with yield (*p*-value < 0.05). Shoot length at the vegetative stage, chlorophyll content of the flag leaf, leaf width at the vegetative stage, root dry biomass, shoot dry biomass, straw weight, and 50% flowering time were also correlated with yield but less significantly (*p*-value < 0.1) (Appendix A).

### 2.3. Partial Factor Productivity (PFP)-Associated Parameters for N/P

The ratio of filled grain yield per unit of applied N/P was expressed as PFP-N or PFP-P for each of the genotypes, and the correlation of significant N/P responsive parameters with PFP was examined. Figure 3 shows the scatter plots of the mean values of each of the parameters from three genotypes at normal, medium, and low doses of N and their correlation with corresponding PFP-N values. Out of 26 N-responsive parameters examined, 20 were significantly correlated with PFP-N (*p*-value < 0.05). In addition, leaf chlorophyll content at the vegetative stage was correlated with PFP-N at a less significant level (*p*-value < 0.1) (Appendix A). Similarly, for P (Figure 4), out of 31 P-responsive parameters, 20 were significantly correlated with PFP-P at a *p*-value of <0.05, while the total number of filled seeds per panicle was less significantly correlated (*p*-value of <0.1) (Appendix A). Out of 19 significantly N- and P-responsive common parameters, 8 parameters were significantly correlated with both yield and PFP-N/P at a *p*-value of <0.05.

### 2.4. Harvest Index-Associated Parameters for N/P

The ratio of grain yield and biomass yield in percentage is expressed as HI for each of the genotypes, and the correlation of significant N/P responsive parameters with HI was examined. Figure 5 shows the scatter plots of the mean values of each of the parameters from three genotypes at normal, medium, and low doses of N and their correlation with corresponding HI values. Out of the 26 N-responsive parameters, 20 were significantly correlated with HI at a *p*-value of <0.05. In addition, the number of green and yellow leaves at maturity and number of unfilled seeds were correlated with HI at a less significant level (*p*-value < 0.1) (Appendix A). Similarly, for P (Figure 6), out of 31 P-responsive parameters, 20 were significantly correlated with HI at a *p*-value of <0.05, while the total number of filled seeds per panicle was less significantly correlated (*p*-value < 0.1) (Appendix A).

Out of 19 significantly N and P-responsive common parameters, 9 parameters were significantly correlated with both yield and PFP at a *p*-value of <0.05. Shoot length at the vegetative stage, shoot biomass, and straw weight were also correlated with yield, but less significantly (*p*-value < 0.1). These three parameters were significantly correlated with HI at a *p*-value of <0.05. Overall, 12 out of 19 parameters were significantly correlated with yield, HI, and PFP at a *p*-value of <0.05.

### 2.5. N/P Dose-Response on All Measured Parameters

Almost all of the measured vegetative parameters increased with an increase in the doses of either N or P (Appendix A). Among the three genotypes, BPT 5204 had the highest average values for 9 out of 18 vegetative parameters followed by Panvel 1 and CR Dhan 301 for the overall N-response. Similarly, BPT 5204 also had the highest average values for 9 out of 18 vegetative parameters followed by CR Dhan 301 and Panvel 1 for the overall P-response. Overall, CR Dhan 301 had the highest average values for 8 out of 11 reproductive parameters for both N/P responses. For the N-response, 5 out of 11 reproductive parameters increased with the N dose, while the weight of spikelets decreased with the N dose. Similarly, 5 out of 11 reproductive parameters increased with the P dose, except 50% flowering time, which decreased with the P dose. Parameters such as panicle length, number and weight of unfilled seeds, number of spikelets, and fertility ratio did not show a dose-wise N/P response in any genotype.

Among physiological parameters, the transpiration efficiency of the flag leaf at the reproductive stage increased with the N dose. The leaf photosynthetic rate and transpiration rate at the vegetative stage, stomatal conductance of the flag leaf and at the reproductive stage, and the transpiration rate of the flag leaf decreased with an increasing N dose. Similarly, the leaf transpiration efficiency at the vegetative and reproductive stages and internal water use efficiency (internal WUE) of the flag leaf and at the reproductive stage showed a direct relationship with the P dose, while the leaf transpiration rate at the vegetative and reproductive stages, along with stomatal conductance and the transpiration rate of the flag leaf, had an inverse relationship with the P dose. The average values of each parameter for all the genotypes for all the treatments are provided in Appendix A.

### 2.6. Genotype-Dependent Variation in N-Response and NUE

The variation in the magnitude of the N-response and use efficiency among the three genotypes in all 46 parameters at each dose of N was calculated as the coefficient of variation [(Standard deviation/mean) × 100] for each nutrient (Appendix A). The coefficients of variation for 18 vegetative parameters at all doses of N ranged from 1 to 65 percent and for 11 directly measured reproductive parameters from 2 to 66 percent, while 15 physiological parameters varied from 1 to 54 percent. This indicates that the phenotypic variation between genotypes increased developmentally from the vegetative to the reproductive stage. Converting the % values of variance in individual parameters into relative values through pie charts for each N dose revealed that 9 of the 18 vegetative phenotypic traits accounted for over 77% of all the variation in N-response among genotypes (Appendix A). Out of these nine, six traits were common across N doses, of which five vegetative traits were significantly N-responsive and correlated significantly with yield, HI, and PFP-N and accounted for the maximum phenotypic variance among genotypes. These traits are culm thickness, root–shoot ratio, shoot dry biomass, shoot length at the vegetative stage, and straw weight. This shortlisting of 5 out of 18 traits is important for germplasm screening, as the number of genotypes that can be phenotypically evaluated is often limited by the number of traits to be monitored.

Similarly, 6 of the 11 directly measured reproductive traits accounted for over 74% of all the variation in the N-response among genotypes. Out of these six, three traits were common across N doses, of which the weight of unfilled seeds was the only reproductive trait with high phenotypic variance among the genotypes that was significantly N-responsive and correlated significantly with yield, HI, and PFP-N. For physiological parameters, it revealed that 8 of the 15 measured accounted for over 76% of all the variation in the physiological N-response among genotypes (Appendix A). Out of these, the transpiration rate of the flag leaf was the only physiological parameter that was significantly N-responsive and correlated significantly with yield, HI, and PFP-N.

An analysis of the relative contribution of PFP-N and HI to the coefficient of variation between genotypes revealed that while both contributed nearly equally at low N, variability in HI increased at the cost of PFP-N variability through medium to normal N. As vegetative growth is the main differentiator between the two parameters, this means that high genotypic variability in vegetative growth translates into corresponding grain yield and PFP with relatively lower variability. This is consistent with the dose–response data for the vegetative and reproductive parameters in each of the genotypes (Appendix A). These results also revalidate the importance of using multiple indices to determine NUE during germplasm screening or crop improvement.

### 2.7. Genotype-Dependent Variation in P-Response and PUE

The coefficients of variation for 18 vegetative parameters at all doses of P ranged from 1 to 55 percent and for 11 directly measured reproductive parameters from 10 to 66 percent, while 15 physiological parameters varied from 1 to 44 percent. This indicates that phenotypic variation in the P-response among genotypes increased developmentally from the vegetative to the reproductive stage. Converting the % values of variance in individual parameters into relative values through pie charts for each P dose revealed that 9 of the 18 vegetative phenotypic traits accounted for over 69% of all the variation in the P-response among genotypes (Appendix A). Out of these nine, six traits were common across P doses, of which five vegetative traits were significantly P-responsive and correlated significantly with two out of three PUE indices, namely yield, HI, and PFP-P, as well as accounting for the maximum phenotypic variance among genotypes. These traits are culm thickness, the root–shoot ratio, shoot dry biomass, straw weight, and shoot length at the reproductive stage. This shortlisting of 5 out of 18 traits is important for germplasm screening, as the number of genotypes that can be phenotypically evaluated is often limited by the number of traits to be monitored.

A similar analysis for reproductive parameters revealed that 6 of the 11 directly measured traits accounted for over 70% of all the variation in the P-response among genotypes. Out of these six, five traits were common across P doses, of which only three reproductive traits were significantly P-responsive and correlated significantly with yield, HI, and PFP-P. The traits are the weight of spikelets, the weight of filled seeds, and the total panicle weight. For physiological parameters, it was revealed that 8 of the 15 measured accounted for over 65% of all the variation in the physiological P-response among genotypes (Appendix A). Out of these eight, three parameters were common across P doses, of which stomatal conductance of the flag leaf was the only physiological parameter that was significantly P-responsive and correlated significantly with HI.

An analysis of the relative contribution of PFP-P and HI to the coefficient of variation among genotypes revealed that while both contributed nearly equally at medium P, variability in HI increased at the cost of PFP-P variability through low to normal P. As P-availability increases from suboptimal towards saturating P-levels, higher genetic variability in vegetative growth and biomass could make the phenotypic differences more discernible for exploitation than reproductive traits for yield, HI, and PFP-P. This is consistent with the dose–response data for the vegetative and reproductive parameters of each of the genotypes (Appendix A). These results also revalidate the importance of using multiple indices to determine PUE during germplasm screening or crop improvement.

### 2.8. Differential Effect of Nitrate and Phosphate on Measured Parameters

The percent effect of N/P on each of the 46 parameters was calculated at medium or low doses as compared to the normal dose for each genotype and was averaged for all genotypes (Appendix A). These averaged values were plotted for each parameter at normal vs. medium doses (Appendix A) and normal vs. low doses (Appendix A). Fifteen out of the eighteen vegetative parameters showed an increase in their values corresponding to an increasing N/P dose with respect to the medium dose, indicating a direct relationship, though the magnitude varied hugely between 0.32% and 34.41%. The most upregulated parameter at the medium N dose was the number of tillers (24.8%) and at the medium P dose was straw weight (34.4%). The most downregulated vegetative parameter at the medium N/P dose was the root–shoot ratio (−107.9%, −46.4%).

The % effect of increasing the N/P dose with respect to the medium dose was positive for seven of the physiological parameters, though the magnitude varied from 1% to 14.01%. The % effect of increasing the N/P dose was negative for six of the physiological parameters by −1.31% to −37.14%. The most upregulated parameter was IWUE (at the reproductive stage) (14%) at the medium N dose and the transpiration efficiency of the flag leaf (10.2%) at the medium P dose. The most downregulated parameter was the transpiration rate of the flag leaf at the medium N dose (−37.1%) and stomatal conductance of the reproductive stage leaf (−21.9%) at the medium P dose.

The % effect of increasing the N/P dose with respect to the medium dose was positive (0.99–29.09%) for seven of the reproductive parameters and negative for five of them (−1.97% to −462.61%). The most upregulated parameter was the fertility ratio (21.4%) at medium N and the number of panicles (29%) at medium P. The most downregulated parameter was PFP at the medium dose for both N/P (−243.6%, −462.6%).

The % effect of increasing the N/P dose with respect to the low dose was positive for 15 out of 18 vegetative parameters (3.91–51.46%), except the root–shoot ratio, with opposite regulation of two others. The most upregulated parameters at the low N dose and low P dose were the culm thickness (51.2%) and leaf width of the flag leaf (23.9%), respectively. The most downregulated vegetative parameter at the low N/P dose was the root–shoot ratio (−26.8%, −53.8%).

The % effect of increasing the N/P dose with respect to the low dose was negative for eight of the physiological parameters (−0.71% to −46.14%) and positive for four of the physiological parameters (7.67–144%). The most upregulated parameter was the transpiration efficiency of the flag leaf at a low N/P dose (24.2%, 144.8%). The most downregulated parameters were the transpiration rate of the flag leaf at a low N dose (−39.3%) and stomatal conductance of the reproductive stage leaf at a low P dose (−46.1%).

All the reproductive parameters except the number of panicles, PFP, and HI showed opposite effects of N/P at normal vs. low doses. The % effects ranged between 1.59 and 40.17% for positive effects and between −5.07% and −1238.2% for negative effects. The most upregulated parameter at the low N/P dose was the number of panicles (19.6%, 40.1%). The most downregulated parameter was PFP at the low dose for both N (−1080.7%) and P (−1238.2%).

### 2.9. Correlation Matrix of Measured Parameters for N/P

The average N/P response value of each parameter in each of the genotypes/doses was used to generate correlation matrices (Appendix A). They were plotted using Minitab v21.4.2 software, and the summary of significantly correlated parameters is provided in Appendix A. It revealed that the highly negatively correlated parameters (r > −0.9) outnumbered the highly positively correlated parameters (r > +0.9) at medium and low N doses, while it was the opposite at a normal N dose.

Dose-wise Venn selections revealed that shoot length at the vegetative stage, the root–shoot ratio, number and weight of filled seeds, panicle length, number, weight of spikelets, and total panicle weight were positively correlated with yield, PFP-N, and HI. Shoot dry biomass and straw weight were highly positively correlated with each other at all doses of N. The number of tillers was highly positively correlated with culm thickness, as was PFP-N with the root–shoot ratio at normal and medium N doses. Similarly, the weight of spikelets was highly negatively correlated with culm thickness as well as with the number of tillers at normal and medium N doses. The root–shoot ratio and shoot length at the reproductive stage were highly positively correlated with each other at a normal N dose, but highly negatively correlated at a medium N dose. The leaf width at the vegetative stage and the leaf photosynthetic rate at the reproductive stage were highly positively correlated at normal and low N doses. Culm thickness and shoot length at the reproductive stage were highly positively correlated, whereas 50% flowering time and shoot length at the vegetative stage were highly negatively correlated at medium and low N doses.

Similarly, the summary of significantly correlated parameters for the P-response is provided in Appendix A. It reveals that the highly positively correlated parameters (r > +0.9) outnumbered the highly negatively correlated parameters (r > −0.9) at normal and low P doses, while it was the opposite at a medium P dose. Dose-wise Venn selections revealed that shoot length at the vegetative and reproductive stages, chlorophyll content of the flag leaf, the root–shoot ratio, number and weight of filled seeds, panicle length, number and weight of spikelets, and total panicle weight were positively correlated with yield, PFP-P, and HI. These use efficiency indices were also positively correlated with physiological parameters such as the photosynthetic rate of the flag leaf and the reproductive stage leaf.

PFP-P and total panicle weight were highly positively correlated with each other as expected at all doses of P. Weight of spikelets was highly positively correlated with chlorophyll content of the flag leaf, as was shoot dry biomass with straw weight at normal and medium P doses. Similarly, the leaf transpiration efficiency at the vegetative stage was highly negatively correlated with panicle length, as was the stomatal conductance of the flag leaf with root length at normal and medium P doses. The photosynthetic rate of the flag leaf was highly positively correlated with the weight of spikelets, as was the number of spikelets with the weight of unfilled seeds at normal and low P doses. The number of spikelets was highly positively correlated with panicle length at medium and low P doses. Similarly, stomatal conductance of the reproductive stage leaf was highly negatively correlated with culm thickness at medium and low P doses.

### 2.10. Segregation of Different Categories of Parameters for N/P

Principal component analysis segregated the measured phenotypic and physiological parameters separately for N and P responses at all doses combined. As much as 88.8% of the cumulative phenotypic variation in the N-response was explained by the first five components based on eigenvalues. The magnitude and direction of the coefficients for the individual variables were examined (Appendix A). The cumulative phenotypic variation explained by the first two PCs was 61.7%. PC1 explained 38.6% of the variation, including the shoot length at the reproductive stage, chlorophyll content of the flag leaf, root–shoot ratio, total panicle weight, PFP, HI, photosynthetic rate, stomatal conductance, and transpiration rate. PC2 explained 23% of the variation, including culm thickness, shoot dry biomass, total number of leaves at maturity, chlorophyll content, leaf width at reproductive stage, number of panicles, 50% flowering time, transpiration efficiency, and IWUE (Figure 7i).

Similarly, 92.7% of the cumulative phenotypic variation in the combined data of all doses of P was explained by the first five components based on eigenvalues. The magnitude and direction of the coefficients for the individual variables were examined (Appendix A). The cumulative amount of phenotypic variation explained by the first two PCs was 64%. PC1 explained 39% of the variation including shoot length at the vegetative and reproductive stages, leaf chlorophyll content at the reproductive stage, number of panicles, total panicle weight, filled seed weight PFP, HI, photosynthetic rate, stomatal conductance, and transpiration rate at the reproductive stage. PC2 explained 25.1% of the variation, including shoot length and leaf width at the vegetative stage, culm thickness, number of tillers, 50% flowering time, number of unfilled seeds, photosynthetic rate, stomatal conductance, and transpiration rate at the vegetative stage. PC2 also contained transpiration efficiency and IWUE-related parameters (Figure 7ii).

### 2.11. Segregation of Contrasting Genotypes for All N/P Dose Responsive Parameters by PLS-DA

The data shown in Figure 8i reveal the contrasting nature of the N-response in CR Dhan 301 relative to the other two genotypes, BPT 5204 and Panvel 1, for all 46 parameters measured under normal and medium N doses. However, at low N, all three genotypes were in contrast to one another, indicating the importance of screening at low N to discern the differences between genotypes. BPT 5204 and CR Dhan 301 were highly contrasting at all N doses, as indicated by their positions in opposite quadrants in the 2D score plot. Though to a lesser degree, BPT 5204 and Panvel 1 were also contrasting at all N doses (Figure 8i).

Figure 8ii reveals the contrasting nature of the P-response in CR Dhan 301 relative to the other two genotypes, BPT 5204 and Panvel 1, for all 46 parameters measured under medium and low P doses. However, at normal P, CR Dhan 301 was in high contrast with Panvel 1 but not with BPT 5204. CR Dhan 301 and Panvel 1 were highly contrasting at all P doses, as indicated by their positions in opposite quadrants in the 2D score plot. Though to a lesser degree, CR Dhan 301 and BPT 5204 were also contrasting at medium and low P doses. All three genotypes showed clear contrast with one another at low P, indicating the importance of screening at low P to discern the differences between genotypes (Figure 8ii).

### 2.12. Ranking of Parameters by Feature Selection Analyses

Feature selection analyses using RandomForest package (Metaboanalyst 6.0 software) enabled the ranking of the 21 parameters listed in Table 1 for NUE (Appendix A). A similar ranking was performed for the 22 parameters listed in Table 2 for PUE (Appendix A) and 12 common to both NUE and PUE (Appendix A). VIPs of these ranked parameters are shown in Figure 9. A comparison of such ranking analysis using other tools such as Weka 3 and VIP score revealed some differences in their ranking pattern. Therefore, Venn selection was used to identify the parameters common to the top half of the rankings by each of the three tools for NUE and PUE and common to both. For NUE, these highly ranked parameters were leaf chlorophyll content, the photosynthetic rate at the reproductive stage, root length, total number of leaves at maturity, and HI. For PUE, these high-ranking parameters were the number of panicles, shoot dry biomass, culm thickness, shoot length, and the photosynthetic rate at the vegetative stage. More importantly, this exercise revealed the common high-ranking parameters for NUE and PUE, which were the photosynthetic rate at the reproductive stage, number of tillers, root–shoot ratio, culm thickness, leaf width of flag leaf, shoot biomass, and straw weight.

### 2.13. Effect of N Doses on Common N/P Nutrient-Use Efficiency Parameters

The effect of N doses on these 12 common nutrient-use efficiency parameters are shown in Figure 10. It showed that there was a positive effect of increasing the N dose on the shoot length at the vegetative and reproductive stages. The effect was clearly visible in CR Dhan 301 at both stages and at the vegetative stage in Panvel 1. BPT 5204 and Panvel 1 showed a smaller increase in shoot length at medium or normal N as compared to low N. At the reproductive stage, the increase in shoot length was higher in BPT 5204 than in Panvel 1. Among the three genotypes, CR Dhan 301 had the highest shoot length followed by BPT 5204 and Panvel 1 at all N doses. The leaf width of the flag leaf increased with an increase in the N dose, except in CR Dhan 301 at the medium N dose. The dose-wise N-response was clearly visible in BPT 5204. The increase in leaf width from medium to normal N was much higher than from low to medium N, indicating the importance of genotypic differences in leaf width with increasing N. Among the three genotypes, Panvel 1 had the highest leaf width of flag leaf followed by CR Dhan 301 and BPT 5204 at all N doses.

Culm thickness and number of tillers increased with an increase in the N dose, except tiller number in CR Dhan 301 at the medium N dose. Among the three genotypes, BPT 5204 had the highest culm thickness and number of tillers followed by Panvel 1 and CR Dhan 301 at all N doses. Culm thickness was higher in BPT 5204 owing to the high number of tillers. Shoot dry biomass and straw weight increased with an increasing N dose, except in CR Dhan 301 at the medium N dose. The positive effect of the N dose on these two parameters was clearly visible in Panvel 1, unlike in CR Dhan 301. The increase in biomass from low to medium or normal N dose was higher in BPT 5204, compared to the small increase from medium to normal N dose. Among the three genotypes, BPT 5204 had the highest shoot dry biomass and straw weight followed by Panvel 1 and CR Dhan 301 at all N doses. BPT 5204 had higher shoot dry biomass owing to the higher number of tillers and total number of leaves.

The root–shoot ratio showed an inverse relationship with N dose, except in CR Dhan at a low N dose and in BPT 5204 at a normal N dose. CR Dhan 301 at medium and normal N doses maintained less shoot dry biomass but much more root dry biomass, resulting in its high root–shoot ratio. Panvel 1 and BPT 5204 showed a proportionate increase in shoot and root dry biomass with N dose, resulting in a proportionate increase in the root–shoot ratio with dose. Among the three genotypes, CR Dhan 301 had the highest root–shoot ratio (*w*/*w*) followed by Panvel 1 and BPT 5204 at all N doses.

The photosynthetic rate at the reproductive stage increased from medium to normal N doses, though higher photosynthetic rates were maintained in all three genotypes at a low N dose, indicating a non-linear response. In BPT 5204, leaf chlorophyll content at the reproductive stage also increased from medium to normal N doses, resulting in a higher photosynthetic rate at a normal N dose. The number of panicles also increased with an increase in the N dose in all three genotypes, except in Panvel 1 at a medium N dose and in CR Dhan 301 at a low N dose. The dose-wise increase in the number of panicles was relatively sharper in BPT 5204. Time taken for panicle emergence in 50% of the plants (or 50% flowering time) increased with increasing N dose in all three genotypes. The delay in flowering by increasing N, especially from low to normal doses, was higher in BPT 5204 (13 days), followed by Panvel 1 (10 days) and CR Dhan 301 (6 days). This is consistent with the relatively higher biomass accumulation in BPT5204 relative to the other genotypes.

Grain HI (%) showed an inverse relationship with N dose in all three genotypes, except in BPT 5204 at a low N dose and in CR Dhan 301 at a medium N dose. HI was highest in CR Dhan 301 followed by BPT 5204 and Panvel 1 at normal N, whereas Panvel 1 replaced BPT 5204 at medium and low N doses.

### 2.14. Effect of P Doses on Common N/P Nutrient-Use Efficiency Parameters

Similarly, the effect of P doses on these 12 common nutrient-use efficiency parameters are shown in Figure 11 and it showed that there was a small positive effect of increasing the P dose on shoot length at the vegetative and reproductive stages. The effect was most prominent in Panvel 1 at the vegetative stage and to a lesser extent in BPT 5204 at the reproductive stage. Among the three genotypes, CR Dhan 301 had the highest shoot length followed by BPT 5204 and Panvel 1 at all P doses. The leaf width of the flag leaf increased with an increase in the P dose, except in BPT 5204 at a low P dose and in Panvel 1 at a normal P dose. The dose–response pattern in leaf width was more prominent from medium to normal doses in CR Dhan 301 and BPT 5204 than from low to medium doses, whereas in Panvel 1, leaf width peaked from low to medium doses and declined thereafter. These differences between the nutrient responses of different associated parameters in terms of genotypes/stages/doses could be of significant interest for crop improvement.

Culm thickness and number of tillers remained stable from low to medium P doses in all three genotypes with even a slight decrease in BPT 5204. However, both parameters increased from medium to normal P doses, more prominently in Panvel 1 than in BPT 5204, while CR Dhan 301 remained stable. Among the three genotypes, BPT 5204 had the highest culm thickness and number of tillers followed by Panvel 1 and CR Dhan 301 at all P doses. Culm thickness was higher in BPT 5204 owing to the higher number of tillers.

Shoot dry biomass and straw weight increased with increasing P dose, more prominently in Panvel 1 and BPT 5204, but not in CR Dhan 301 at a low P dose. The increase in biomass from the low to medium P dose was higher, whereas it was small from the medium to normal P dose in Panvel 1, indicating saturation. CR Dhan 301 maintained higher biomass at a low P dose, declined with the increase in P to a medium dose, and remained stable thereafter at the normal dose. Among the three genotypes, BPT 5204 had the highest shoot dry biomass and straw weight, followed by Panvel 1 and CR Dhan 301 at all P doses, except in CR Dhan 301 at a low P dose. BPT 5204 had higher shoot dry biomass owing to the higher number of tillers and total number of leaves.

The root–shoot ratio (*w*/*w*) showed an inverse relationship with the P dose, except in CR Dhan 301 at a low P dose. At medium and normal P doses, CR Dhan 301 maintained higher biomass in the roots than shoots, resulting in its high root–shoot ratio, whereas it was the opposite at a low dose, resulting in a smaller root–shoot ratio. CR Dhan 301 had the highest and BPT 5204 had the lowest root–shoot ratio at normal and medium P doses.

The photosynthetic rate increased with the P dose at the reproductive stage only in BPT 5204. The increase from a low to medium P dose was higher than from a medium to normal P dose, indicating saturation. The pattern was opposite in CR Dhan 301 and Panvel 1. In BPT 5204, leaf chlorophyll content at the reproductive stage also increased with the P dose, resulting in a higher photosynthetic rate at a normal P dose. The number of panicles also increased with an increase in P dose in all three genotypes, except in BPT 5204 at the medium P dose and in CR Dhan 301 at the normal P dose. In CR Dhan 301, the number of panicles increased from low to medium P doses and remained stable thereafter. The dose-wise increase in the number of panicles was relatively sharper in Panvel 1. The number of days for panicle emergence in 50% of the plants (or 50% flowering time) was significantly delayed from medium to normal P doses in CR Dhan 301 (6 days), while it was slightly delayed in Panvel 1 (2 days). In BPT 5204, an increase in dose from medium to normal P promoted faster flowering (5 days). For all three genotypes, the 50% flowering time was very delayed at low P when compared to medium and normal P doses, indicating the importance of the presence of an adequate amount of P for timely panicle emergence and flowering.

Grain HI (%) showed an inverse relationship with P dose in BPT 5204 and Panvel 1, while in CR Dhan 301, it peaked at medium P dose and declined thereafter. HI was highest in CR Dhan 301 followed by BPT 5204 and Panvel 1 at normal P, but at medium and low P doses, Panvel 1 replaced BPT 5204.

### 2.15. Effect of N/P Doses on Yield, Partial Factor Productivity, and Harvest Index

The effect of N/P doses on all 3 nutrient use efficiency indices, namely yield, PFP and HI is shown in Figure 12. At low N, Panvel 1 and CR Dhan 301 had the highest yield, PFP-N, and HI, in sharp contrast with BPT 5204, indicating considerable scope for its improvement towards NUE. At medium and normal N doses, CR Dhan 301 distinctly contrasted with the other two genotypes with the highest yield and HI, though PFP-N fell sharply at a normal N dose. In other words, the retention of the contrastingly high yield and HI in CR Dhan 301 at normal N was achieved at the cost of PFP-N, indicating the diversion of excess N into shoot biomass rather than grain yield.

For the P-response, CR Dhan 301 had the highest yield, PFP-P, and HI at all P doses and distinctly contrasted with the other two genotypes, though PFP-P fell sharply at a normal P dose. In other words, the retention of the contrastingly high yield and HI in CR Dhan 301 at normal P was achieved at the cost of PFP-P, indicating the diversion of excess P into shoot biomass rather than grain yield. Based on all three NUE/PUE indices, CR Dhan 301 showed a sharp contrast with the other two genotypes, BPT 5204 and Panvel 1, regarding the N/P response.

Yield was calculated as the weight of filled seeds per panicle in this study. It increased with the increase in N/P doses in BPT 5204 and CR Dhan 301, while it was the opposite in Panvel 1. The weight of filled seeds was higher at a low N dose, owing to the higher number of filled seeds at a low N dose in Panvel 1. Among the three genotypes, CR Dhan 301 had the highest yield followed by BPT 5204 and Panvel 1 at all doses of N/P, except Panvel 1 at a low N dose.

Partial factor productivity showed an inverse relationship with N/P doses in all three genotypes. PFP-N was highest in CR Dhan 301 followed by BPT 5204 and Panvel 1 at normal N, while at low N, Panvel 1 had the highest PFP-N followed by CR Dhan 301 and BPT 5204. The extent of the increase in PFP-N at low N from normal N was approximately 85% in both BPT 5204 and CR Dhan 301, whereas in Panvel 1, it was 95% owing to the increased filled seeds number, yield, and total panicle weight at a low N dose. PFP-P was the highest in CR Dhan 301 followed by BPT 5204 and Panvel 1 at all P doses. The extent of the increase in PFP at low P from normal P was approximately 91% in both BPT 5204 and Panvel 1 and 94% in CR Dhan 301.

The grain harvest index (HI, %) showed an inverse relationship with N dose in all three genotypes, except in BPT 5204 at a normal N dose and in CR Dhan 301 at a medium N dose. HI was the highest in CR Dhan 301 followed by BPT 5204 and Panvel 1 at normal N. Panvel 1 had higher HI than BPT 5204 at medium and low N doses. The extent of the increase in HI at low N from normal N dose was approximately 34% and 4% in BPT 5204 and CR Dhan 301, respectively. There was a 73% increase in Panvel 1 HI at a low N dose from a normal N dose owing to the increased PFP-N, filled seeds number, filled seeds weight, and total panicle weight at low N as compared to the normal N dose.

Grain HI also showed an inverse relationship with P dose in BPT 5204 and Panvel 1, except BPT 5204 at a medium P dose, while in CR Dhan 301, it peaked at a medium P dose and declined thereafter. HI was highest in CR Dhan 301 followed by BPT 5204 and Panvel 1 at normal P. Panvel 1 had a higher HI than BPT 5204 at medium and low P doses. The extent of the increase in HI at low P from the normal P dose was approximately 10% and 47% in BPT 5204 and Panvel 1, respectively, whereas it was 7% at medium P from the normal P dose in CR Dhan 301.

### 2.16. Validation of Shortlisted NUE/PUE Candidate Genes Using RT-PCR

A Venn selection was carried out using 389 NUE-related candidate genes [22] and 103 PUE-MQTL-associated genes [45] in rice using the Venny 2.1 tool. This revealed five common genes for NUE and PUE. Their differential gene expression at each of the three N/P doses was examined by qRT-PCR in all three genotypes, using the primers provided in Appendix A and actin as an internal housekeeping control.

*OsFER2* encodes ferritin, an iron storage protein. There was a N dose-dependent decrease in its expression in Panvel 1 relative to low N (Figure 13). In CR Dhan 301, it was highly downregulated at the medium N dose and less downregulated at the normal N dose, and in BPT 5204, it was downregulated at both doses. *OsEXPA10* encodes the expansin protein involved in root cell elongation. Its expression relative to low N was unaffected at the medium N dose but downregulated at the normal N dose in Panvel 1. It was highly upregulated at both doses in BPT 5204 and highly downregulated at both doses in CR Dhan 301. *OsSultr3;4* encodes the SULTR-like phosphorus distribution transporter involved in the control of P allocation to the grain. It showed the opposite pattern in BPT 5204 (downregulated) and CR Dhan 301 (upregulated) at both N doses, while in Panvel 1, it showed upregulation at a medium N dose and downregulation at a normal N dose. *OsCYP75B4* encodes Chrysoeriol 5’-Hydroxylase involved in Tricin biosynthesis. Its expression showed dose-dependent downregulation to different extents in all three genotypes. *OsIAA3* is similar to the auxin-responsive protein gene. Its expression was upregulated in both BPT 5204 and Panvel 1 with dose-dependence in BPT 5204, while in Panvel 1, its expression peaked at the medium N dose and declined thereafter. In CR Dhan 301, it was highly downregulated at the medium dose and marginally upregulated at the normal N dose. These data reveal the hitherto unknown N-response of these genes in rice, as well as their genotype-dependent regulation.

Overall, *OsSultr3;4* and *OsEXPA10* showed N-responsive opposite regulation in one pair of genotypes contrasting for NUE (CR Dhan 301 and BPT 5204), while *OsSultr3;4* was oppositely regulated at the medium N dose and *OsIAA3* at the normal N dose in the other contrasting pair (CR Dhan 301 and Panvel 1). These data clearly indicate the importance of these candidate genes as targets to improve NUE.

As far as the P-response of these genes is concerned, *OsFER2* expression was highly downregulated in both BPT 5204 and Panvel 1 genotypes and highly upregulated in CR Dhan 301. It showed a dose-dependent decrease in Panvel 1 but was unaffected at the normal dose and peaked at the medium P dose in BPT 5204. *OsEXPA10* gene expression remained unaffected at the medium dose and highly upregulated at the normal P dose in BPT 5204. In CR Dhan 301, its expression was highly downregulated at the medium dose and remained unaffected at the normal P dose but was marginally downregulated at the medium dose and highly downregulated at the normal P dose in Panvel 1. The expression of the *OsSultr3;4* gene showed high downregulation at the medium dose and upregulation at the normal P dose in CR Dhan 301. In Panvel 1, it showed upregulation at the medium dose and downregulation at the normal P dose, while in BPT 5204, it was unaffected at both P doses. The *OsCYP75B4* gene was downregulated at the medium P dose in BPT 5204 and CR Dhan 301, while it was highly upregulated in BPT 5204 and remained unaffected in CR Dhan 301 at the normal P dose. In Panvel 1, it was highly upregulated at the medium dose and highly downregulated at the normal P dose. *OsIAA3* gene expression was upregulated at the medium P dose and downregulated at the normal P dose in BPT 5204 and CR Dhan 301, although to different extents. In Panvel 1, it was highly downregulated at both doses.

Overall, *OsFER2* showed P-responsive opposite regulation in one contrasting pair for PUE (CR Dhan 301 and BPT 5204), while *OsFER2*, *OsSultr3;4*, and *OsCYP75B4* showed opposite regulation in the other contrasting pair (CR Dhan 301 and Panvel 1). These data clearly indicate the importance of these candidate genes as targets to improve PUE.

## 3. Discussion

Nitrogen and Phosphorus compounds are very important for crop performance, and the challenge of their low use efficiencies has largely been addressed separately in many crops including rice. The main contribution of the present work is to study the N/P response and their use efficiencies together, in terms of the underlying phenotypic, physiological, and gene expression patterns in three popular rice genotypes. They are high-yielding and popular in different Indian agroclimatic zones. There is no literature characterizing the N/P response or efficiency in CR Dhan 301 and only one report each on N/P in BPT5204 [46,47]. Panvel 1 has been well studied for N-response/NUE [23,24,27,46,48], but to a far lesser extent for P [49]. This is also by far the most comprehensive evaluation of 46 phenotypic and physiological parameters for N/P response and use efficiency, coupled with the analysis of 5 genes common to both NUE and PUE in any crop.

### 3.1. Morpho-Physiological Characterization of NUE and PUE

This study has revealed, for the first time, eight vegetative traits, three reproductive traits including HI, one physiological trait, and five genes as common to NUE and PUE. In addition, there are seven vegetative traits that are common to the N/P response, even though there is no evidence of their contribution to N/P use efficiencies at present, either in our study or in the literature. While there are no such studies on traits common to the N/P response or use efficiency in rice, there have been some reports on other crops such as maize [50] and a compilation of studies on wheat [51]. Subject to field evaluation, these common traits for NUE/PUE could be of immense interest for crop improvement to bring together multiple nutrient use efficiencies and maximize their economic, agronomic, and environmental benefits for the sustainability transition, at least in rice. Some molecular studies have reported genes common to the N/P response [52,53,54], but none for both NUE and PUE so far, though few have studied their QTLs separately [45,55].

These three genotypes have not figured into any ranking earlier, except our ranking of Panvel 1 as high-NUE among other genotypes [23,24]. Here we report CR Dhan 301 to have higher NUE among these three genotypes at medium and normal N doses, apart from the highest PUE at all P doses. It is only at a low N dose that Panvel 1 had the highest NUE.

We shortlisted 21 out of the 46 measured parameters as NUE parameters. These include 13 vegetative, 3 physiological, and 5 reproductive parameters. Several of these vegetative traits including chlorophyll content, shoot length (plant height), plant dry biomass, root length, and number of leaves and tillers were already known to be associated with NUE [23,27,48,56]. Other traits hitherto unreported for NUE and found here are culm thickness, leaf width of flag leaf, number of yellow leaves, and the root–shoot ratio. Similarly, the photosynthetic rate, number of panicles, and 50% flowering time were also known as NUE parameters [24,57,58], but the transpiration rate and efficiency in flag leaf were found to also contribute to NUE in this study.

The general pattern of the N-response for all these parameters was consistent among all three genotypes, even in terms of NUE parameters, except that Panvel 1 had higher yield and PFP at low N vs. normal N dose. These findings are more consistent across our own results and with the literature when we compare the normal dose with any lower dose, whether the medium or low dose, as commonly used with one or more of the same genotypes [23,24,27,46,48,58]. These similarities indicate the consistency in our results, even as we advance them with more treatments/parameters/genotypes/nutrients.

It was reported that in 30-day-old plants, BPT 5204 had a higher photosynthetic rate, stomatal conductance, chlorophyll content, photosynthetic NUE, and yield than the Panvel 1 genotype, though all the observations were not statistically significant [46]. However, in plants 45 days or older, we found that Panvel 1 had a higher yield and PFP-N than BPT 5204 with more robust statistical significance. The low genotypic differences for photosynthetic rate in our study could be due to less light in the greenhouse relative to field conditions, though most other measured parameters did reveal significant genotypic differences (Appendix A). Such differences were also reported using other genotypes in rice [48,58].

We found that all the parameters that showed a high degree of genotype-dependent variation did not necessarily rank highly in terms of their correlation with NUE, perhaps because all correlations may not have a causal basis. On the other hand, some of our highly ranked parameters with small but significant genotypic differences were consistent with the literature concerning other genotypes, such as chlorophyll content and the photosynthetic rate at the reproductive stage, though they were not ranked. This was true for greenhouse studies [46] as well as field studies [48,58], indicating that our findings could be field-relevant, subject to further validation.

In relation to P, the main emphasis in the literature has been the P-starvation response rather than the P dose–response pattern and PUE. We shortlisted 22 out of 46 measured parameters as PUE parameters, spanning 10 vegetative, 6 physiological, and 6 reproductive parameters. Several of these vegetative parameters were known to be associated with PUE such as shoot length (plant height), shoot dry biomass, the root–shoot ratio, and number of tillers [59,60,61]. In addition, we contributed culm thickness, leaf width of the flag, and reproductive leaf as PUE traits here. Similarly, the photosynthetic rate, number of panicles, and 50% flowering time were previously associated with PUE [62,63]. In addition, we contributed stomatal conductance, leaf transpiration rate at the vegetative stage, leaf internal WUE at the reproductive stage, and weight of spikelets as PUE parameters in this study.

The general pattern of P-response for all these parameters was consistent among all three genotypes, even in terms of PUE parameters. These findings are more consistent across our own results and with the literature when we compare the normal dose with any lower dose, whether medium or low doses, as commonly used with one or more of the same genotypes [47,49,59,60,64]. These similarities indicate the consistency in our results, even as we advance them with more treatments/parameters/genotypes/nutrients.

The effects of a low P dose observed in our study were reduced shoot and root growth (length), decreased number of leaves, leaf size reduction in terms of width, less chlorophyll content, more prominent decrease in shoot biomass than the root biomass, and an increased root–shoot biomass ratio.

Interestingly, we found that, unlike in NUE, there was a better correspondence between the correlation-based ranking for PUE and the extent of genotypic variation in those parameters, indicating an underlying biological basis. They were also consistent with the genotypic variations reported in other genotypes for shoot length/plant height and shoot/total dry biomass, though unranked for PUE, whether in the lab [49,59,60] or field studies [64]. This indicates that our findings on PUE could be field-relevant, subject to further validation.

### 3.2. Candidate Genes for NUE and PUE

Genetic improvement for these quantitative traits had been hampered up until a decade ago by the limited availability of phenotypic or genomic data, making it difficult to develop markers for selection/breeding [2,65]. However, things have changed with recent advances, especially in rice [14], in terms of phenotyping [23,24,66,67], germplasm evaluation [68,69,70,71], and functional genomics and bioinformatics [22,25,27,28,45,72]. Nevertheless, further work is required to identify the minimal common parameters for NUE and PUE, as genome-wide associations still rely on different parameters [59,73,74,75]. The identification of common traits and genes for NUE and PUE shall open tremendous opportunities for crop improvement by selection and/or breeding.

We found significant genotype-dependent differential expression of five genes common to NUE and PUE for the first time. *OsFER2* (Os12g0106000) is an iron storage protein in chloroplast known to be induced by N starvation in rice roots [76]. We found that it was highly downregulated with increasing N doses in all three genotypes, indicating an antagonistic interaction between N and Fe in rice. *OsEXPA10* (Os04g0583500), a cell wall-loosening protein, is induced by Al and regulates root cell elongation involving the ART1 transcription factor in rice [77]. It is associated with plant height, grain length, and root development traits in rice [78] and was linked to NUE in our previous studies [28]. It was upregulated in BPT 5204 and downregulated in CR Dhan 301, indicating its negative regulation in the high-NUE genotype, CR Dhan 301.

The *OsSultr3;4* (Os06g0143700) gene is a SULTR-like phosphorus distribution transporter, which was reported to be a G-protein (RGA1)-regulated gene in rice [79]. Its upregulation in CR Dhan 301 by the N dose indicates that it could be a positive regulator of NUE. *OsCYP75B4* (Os10g0317900) is a cytochrome P450 75B4 gene known to be upregulated under N deficiency in rice [76]. It is highly downregulated in all three genotypes, indicating its negative regulation in NUE. The *OsIAA3* (Os01g0231000) gene codes for an auxin-responsive protein known to be downregulated in N deficiency and associated with grain length in rice [76]. Its upregulation in BPT 5204 and Panvel 1 and downregulation in CR Dhan 301 indicate that it regulates NUE through different mechanisms in contrasting genotypes. Overall, these data reveal the hitherto unknown N-response of these five genes in rice, as well as their genotype-dependent regulation.

With regard to P, the *OsFER2* (Os12g0106000) gene is also known to be highly upregulated by P starvation in rice [52]. The antagonistic interaction between Fe and P is well known, resulting in increased Fe concentrations in the root cells during P deficiency conditions. This gene was also manipulated to improve the nutritional quality of rice grains [80]. At a normal P dose, it showed high upregulation in the high-PUE genotype, CR Dhan 301, indicating opposite regulation at both normal doses of N and P. The *OsEXPA10* gene is associated with P-sensitivity in rice [78]. Its high downregulation at a medium P dose indicates its negative regulation for PUE.

The *OsSultr3;4* gene is known to be associated with phosphorus content, cold, and salt tolerance traits in rice [78]. Genetic manipulation of this gene results in different levels of phosphorus in grains and leaves in rice [81]. It is highly downregulated at a medium P dose, indicating it to be a negative regulator of PUE. The *OsCYP75B4* gene showed high downregulation at a medium P dose in CR Dhan 301, indicating that it could be a negative regulator of PUE in rice. The differential gene expression of the *OsIAA3* gene in contrasting genotypes indicates it could also be an important regulator of PUE in rice. These five genes were highly upregulated under P deficiency conditions in rice [45]. These data reveal the hitherto genotype-dependent regulation of these five genes in rice.

Overall, these five genes showed opposite N/P regulation or large changes in expression levels (Figure 13) between contrasting genotypes, namely CR Dhan 301 and BPT 5204/Panvel 1 of NUE/PUE (Figure 8 and Figure 12). This clearly indicates their role in NUE/PUE and indicates them as candidate genes for further validation.

In conclusion, we report the genotype-dependent differential expression of five genes common to those suggested separately in the literature for NUE [22] and PUE [45], apart from comprehensively characterizing the phenotypic parameters for both traits together for the first time. Our results reveal a strong correspondence between phenotypic traits and differential gene expression between contrasting genotypes and the potential of these five candidate genes to improve both NUE and PUE.

## 4. Materials and Methods

### 4.1. Plant Material and Growth Conditions

The three genotypes selected for the present study were CRDhan 301, Panvel-1, and Samba Mahsuri (BPT-5204). CR Dhan 301 is a high-yielding, medium-duration variety popular in Odisha in Eastern coastal India, with long slender grain and moderate tolerance to biotic and abiotic stress, (http://www.rkbodisha.in). Panvel-1 is also a high-yielding, medium-duration variety with high salt tolerance popular in Western coastal India (http://aicrp.icar.gov.in). BPT 5204 is a high-yielding, long-duration variety widely cultivated across South India and popular for its excellent cooking quality (http://aicrp.icar.gov.in). Their seeds were procured from the Kharland Rice Research Station, Panvel, Maharashtra, the National Rice Research Institute (ICAR-NRRI), Cuttack, Orissa, the Indian Institute of Rice Research (IIRR), Hyderabad, and the ICAR-Indian Agricultural Research Institute (IARI), New Delhi. Individual seeds of these genotypes were weighed, and seeds of modal weight (±1 mg) were used for greenhouse experiments as described earlier [23].

### 4.2. Greenhouse Growth Conditions

The seeds of the three genotypes, CR Dhan 301, Panvel-1, and BPT-5204, were germinated in square Petri plates on sterile moist cotton containing Arnon-Hoagland media [82] with different doses of N/P in a plant growth chamber at 28 °C, 270 µmolm^−2^ s^−1^ light intensity, 70% humidity, and a 12:12 h (light:dark) photoperiod. The three nitrogen doses used were 10% N (1.5 mM nitrate, control), 50% N (7.5 mM), and 100% N (15 mM). The three phosphorus doses used were 5% P (0.05 mM phosphate, control), 10% P (0.1 mM), and 100% P (1 mM). After 10 days of growth (from the date of plating), the seedlings for each treatment were transferred into 25 replicate pots (2 seedlings per pot) containing nutrient-depleted soil [83] supplemented with modified AH media as above. The pots were placed in the greenhouse in a randomized block design and supplemented with media (40 mL per pot) every 2–3 days as needed to keep the soil moist.

### 4.3. Vegetative Growth Measurements

The greenhouse-grown plants were monitored for growth-related parameters as per IRRI-IBPGR [24] in 20 replicates per treatment at the vegetative stage, 50 days after transplantation in the soil (DAT), and/or at the reproductive stage (95 DAT). Shoot length was measured in cm using a measuring tape. Chlorophyll content was measured in fully expanded leaves (2 leaves per plant) using a Chlorophyll meter (SPAD 502 DL PLUS, Konica Minolta, Japan) and recorded as SPAD values. At the reproductive stage (95 DAT), chlorophyll content was measured in both the flag leaf and the one below it. Leaf width was measured in cm from the middle position of the leaf, using a measuring scale. Stem thickness was measured once after 95 DAT from the base of the plant using Vernier calipers (in mm). The number of tillers per plant was counted at the end of tillering stage of each genotype/treatment. Root length was measured in cm using a measuring tape after harvest. The root–shoot ratio was calculated in terms of root and shoot biomass [60]. The number of green leaves, yellow leaves, and total leaves at maturity were counted manually.

### 4.4. Physiological Measurements

Photosynthetic and transpiration-related parameters were measured using the LI-6400/XT portable photosynthesis system (LI-COR Biosciences, Lincoln, NE, USA). The net assimilation rate, Pn (µmol CO_2_ m^−2^ s^−1^), stomatal conductance, gs (mol H_2_O m^−2^ s^−1^), and transpiration rate, Tr (mmol m^−2^ s^−1^), were measured in the fully expanded uppermost leaves at the vegetative (50 DAT) and reproductive (95 DAT) stages [25]. Transpiration efficiency was measured in terms of μmol (CO_2_)/mmol (H_2_O) m^−2^s^−1^ [84] and internal WUE (Internal WUE) was measured in terms of μmol CO_2_/mol (H_2_O) [85]. The reference CO_2_ concentration was 400 ± 20 μmol mol^−1^ during the measurements. All LICOR measurements were carried out at the time of maximal photosynthetic activity between 11:00 a.m. and 2:00 p.m. IST. All the measurements were performed in five independent replicates.

### 4.5. Reproductive and Yield-Related Measurements

Days to panicle emergence were recorded in each genotype, and the days to 50% flowering (50% FD) were calculated as the time taken for panicle emergence in 50% of the plants for each genotype/treatment. The number of panicles was measured in each plant after the panicle maturation stage. After maturation, panicles were harvested from each plant to record panicle length (in cm) and total panicle weight (in g/plant). The number of spikelets per panicle was counted. Seeds were separated from the panicle, segregated into filled and unfilled seeds, counted, weighed (in g), and recorded separately. The ratio of filled grains per panicle or the fertility ratio was calculated as the ratio of the number of filled grains per panicle to the total number of spikelets per panicle [86]. Shoot and root tissues were harvested separately, and roots were washed with distilled water and dried and their length was measured in cm using a measuring tape. Shoot fresh weight was recorded in grams. For dry weight, the shoot and root tissues were dried at 65 °C using a hot-air oven until a constant weight was achieved (in g). Partial factor productivity and the harvest index were calculated as per the formulae below [87]:$$Partial\;factor\;productivity\;(PFP-N/P)=\frac{weight\;of\;filled\;seeds\;per\;plant\;(g)}{N/P\;applied\;per\;plant\;(g)}$$
$$Harvest\;Index\;\left(HI\right)=\frac{Total\;grain\;weight\;(g)}{Shoot\;dry\;weight\;\left(g\right)}\times 100$$

### 4.6. RNA Isolation and qRT-PCR Analysis

Leaves from the potted plants of all 3 genotypes grown with media containing normal, medium, or low N/P levels were harvested at the reproductive stage and immediately frozen in liquid nitrogen as 100 mg aliquots. Total RNAs were isolated from them using RNAiso Plus solution and 3 µg each were reverse-transcribed using the PrimeScript^TM^1st strand cDNA synthesis kit as per the instructions of the supplier (Takara, Kusatsu, Japan). Exon-spanning primers were designed using the Quant Prime tool [88]. RT-qPCR was performed using SYBR Green qPCR MasterMix (GBiosciences, St. Louis, MO, USA) and the Aria Mx Real-time PCR System (Agilent technologies, Singapore). The relative abundance of transcripts was calculated using the 2^−ΔΔCT^ method [89] using the actin gene (LOC_Os01g64630) as an internal control. The data were statistically analyzed using GraphPad Prism 6 software. These experiments were performed using two biological and three technical replicates.

### 4.7. Statistical Analyses

Statistical analysis of the data was performed using GraphPad Prism Version 6.01 for Windows (GraphPad Software, Boston, MA, USA, www.graphpad.com). The values were averaged and recorded as mean ± SE. Two-way ANOVA tests were performed on these averaged values of parameters to identify the variations due to doses, genotypes, nutrients, or their interactions. A multiple comparison test was performed using Tukey’s test using *p* ≤ 0.05. The correlation and PCA analyses were performed using Minitab Statistical software, https://www.minitab.com. PLS-DA and ranking by the Randomforest package were performed using MetaboAnalyst 6.0. software, http://www.metaboanalyst.ca [90].

## Figures and Tables

**Figure 1 plants-13-02567-f001:**
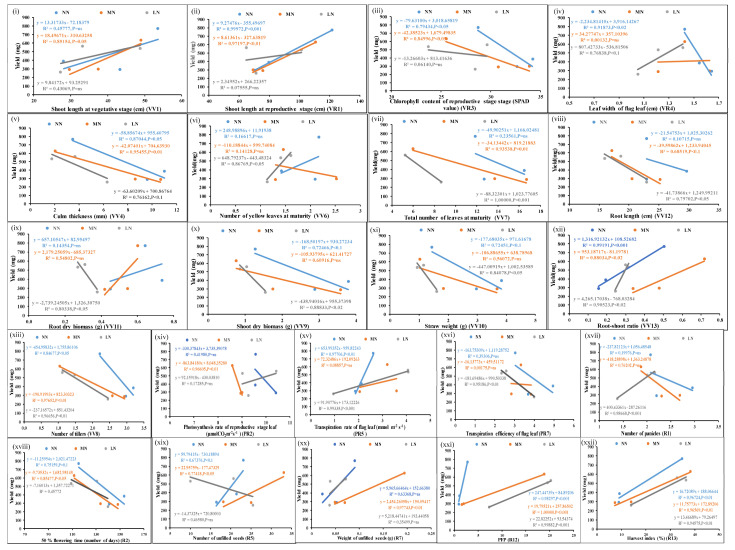
Scatter plots showing significant correlations of yield with N-responsive parameters. Correlations of the mean values of each of the parameters, (**i**–**xxii**) from three genotypes at normal, medium, and low doses of N with corresponding yield values at *p*-values of <0.01, 0.02, 0.05, and 0.001 showing significant correlations; *p*-value of <0.1 means less significant correlation, ns means non-significant correlation.

**Figure 2 plants-13-02567-f002:**
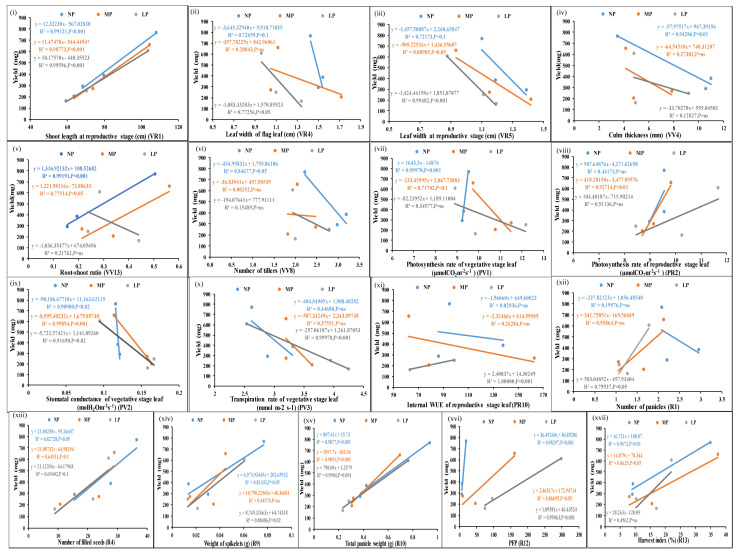
Scatter plots showing significant correlations of yield with P-responsive parameters. Correlations of the mean values of each of the parameters, (**i**–**xvii**) from three genotypes at normal, medium, and low doses of P with corresponding yield values at *p*-values of <0.01, 0.02, 0.05, and 0.001 showing significant correlations; *p*-value of <0.1 means less significant correlation, ns means non-significant correlation.

**Figure 3 plants-13-02567-f003:**
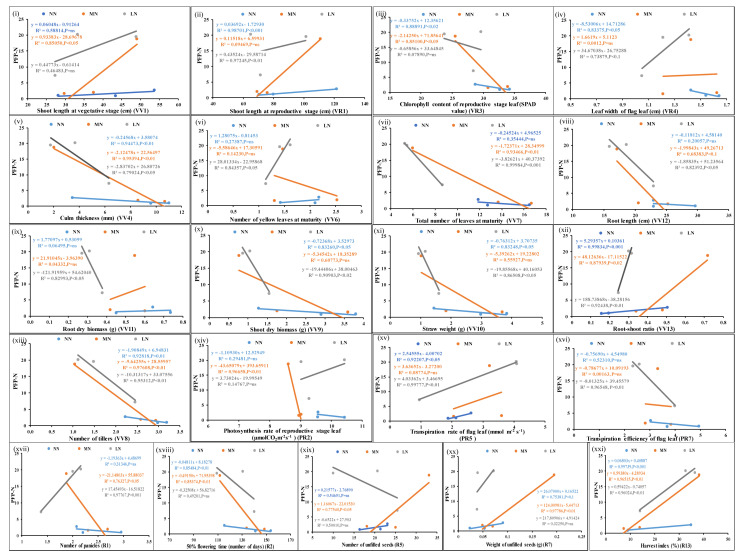
Scatter plots showing significant correlations of PFP-N with N-responsive parameters. Correlations of the mean values of each of the parameters, (**i**–**xxi**) from three genotypes at normal, medium, and low doses of N with corresponding PFP values at *p*-values of <0.01, 0.02, 0.05, and 0.001 showing significant correlations; *p*-value of <0.1 means less significant correlation, ns means non-significant correlation.

**Figure 4 plants-13-02567-f004:**
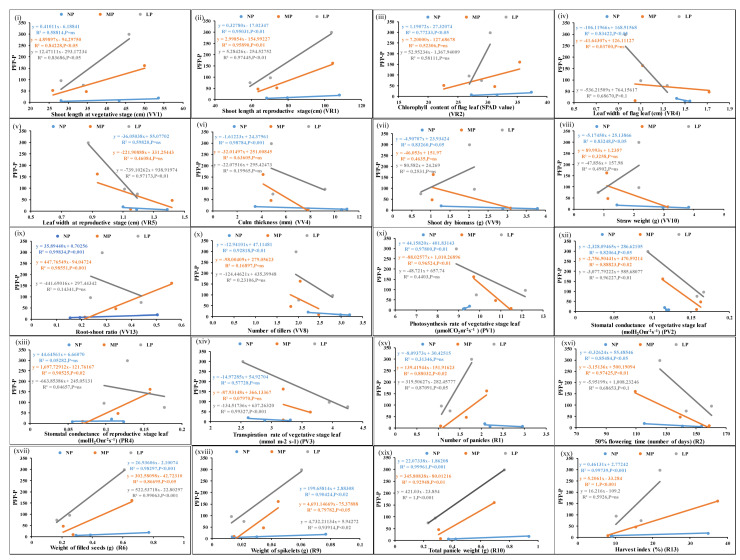
Scatter plots showing significant correlations of PFP-P with P-responsive parameters. Correlations of the mean values of each of the parameters, (**i**–**xx**) from three genotypes at normal, medium, and low doses of P with corresponding PFP values at *p*-values of <0.01, 0.02, 0.05, and 0.001 showing significant correlations; *p*-value of <0.1 means less significant correlation, ns means non-significant correlation.

**Figure 5 plants-13-02567-f005:**
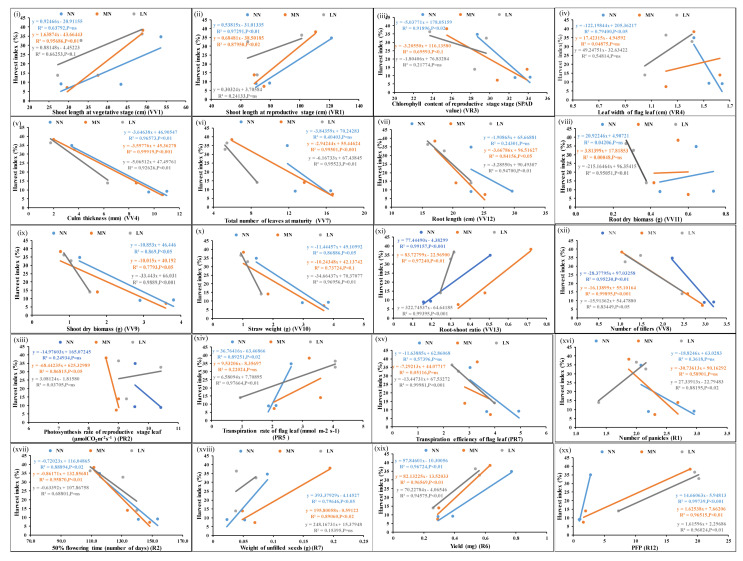
Scatter plots showing significant correlations of HI with N-responsive parameters. Correlations of the mean values of each of the parameters, (**i**–**xx**) from three genotypes at normal, medium, and low doses of N with corresponding HI values at *p*-values of <0.01, 0.02, 0.05, and 0.001 showing significant correlations; *p*-value of <0.1 means less significant correlation, ns means non-significant correlation.

**Figure 6 plants-13-02567-f006:**
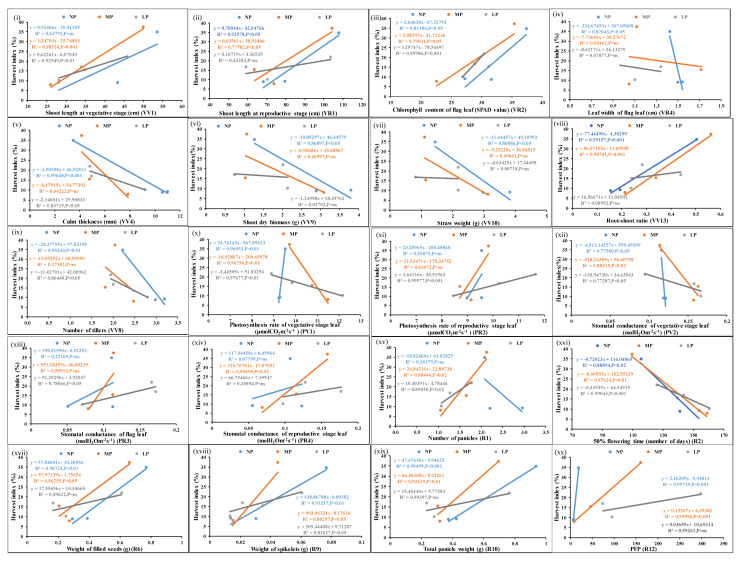
Scatter plots showing significant correlations of HI with P-responsive parameters. Correlations of the mean values of each of the parameters, (**i**–**xx**) from three genotypes at normal, medium, and low doses of P with corresponding HI values at *p*-values of <0.01, 0.02, 0.05, and 0.001 showing significant correlations; *p*-value of <0.1 means less significant correlation, ns means non-significant correlation.

**Figure 7 plants-13-02567-f007:**
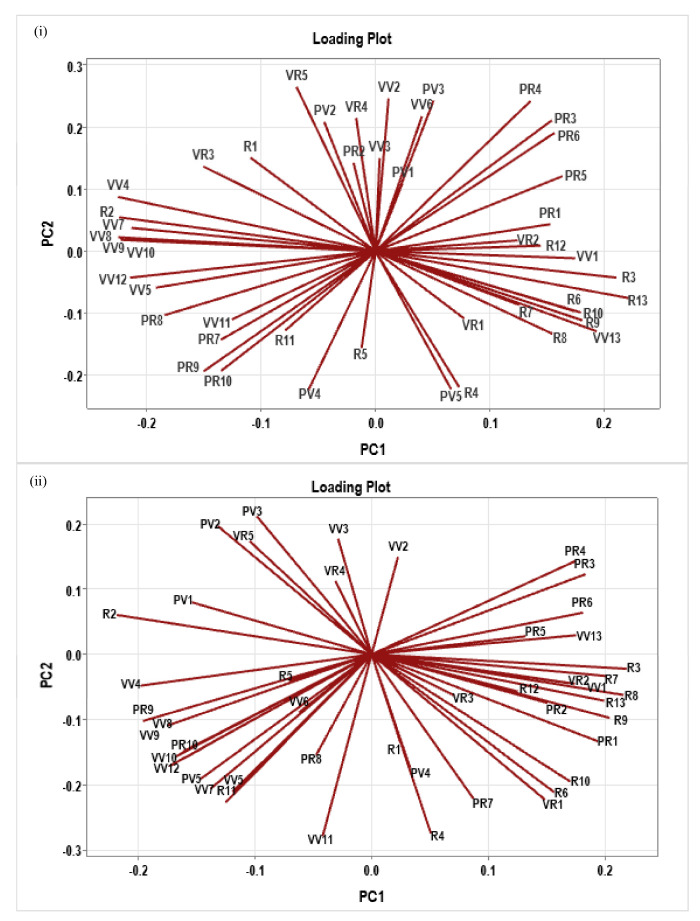
Loading plots of first two principal components of N-response (**i**) and P-response (**ii**). PCA was performed using the data of all parameters at all doses of N and P combined, plotted using Minitab v21.4.2 statistical software.

**Figure 8 plants-13-02567-f008:**
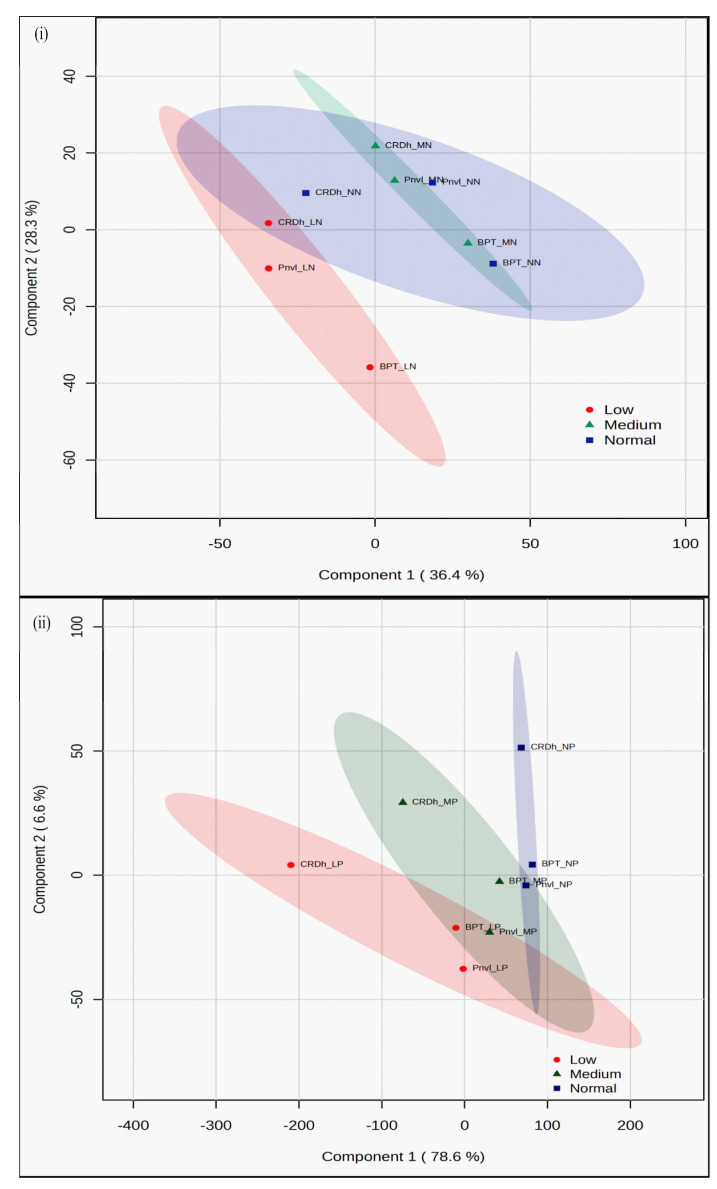
PLS-DA 2D score plots of the N (**i**) and P (**ii**) response of three genotypes (BPT for BPT 5204, Pnvl for Panvel 1, and CRDh for CR Dhan 301). PLS-DA was performed for all 46 measured parameters at 3 doses using MetaboAnalyst 6.0 software.

**Figure 9 plants-13-02567-f009:**
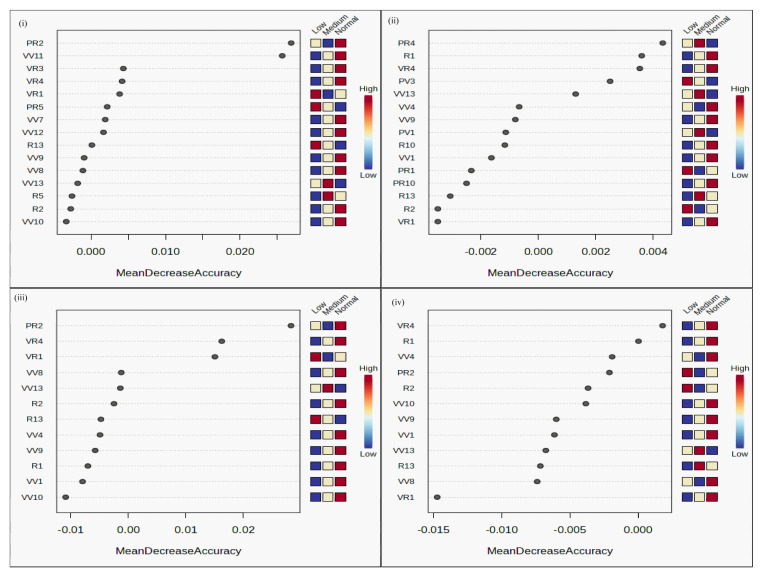
Variable Importance plots showing ranking of parameters. Plots for NUE (**i**) and PUE (**ii**) parameters as analyzed by RandomForest feature selection tool using MetaboAnalyst 6.0 software. The common parameters for NUE and PUE were plotted separately (**iii**,**iv**), in view of the different average values for N/P-response in the respective parameters.

**Figure 10 plants-13-02567-f010:**
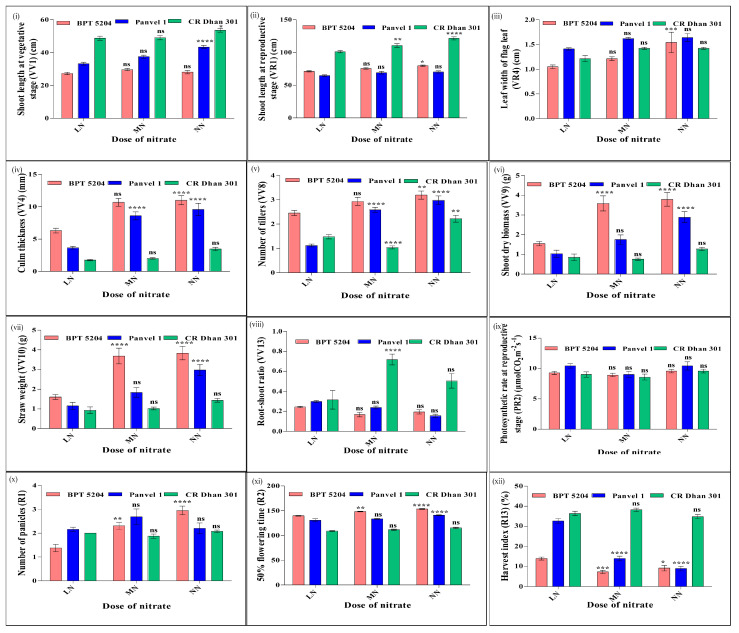
N dose-wise response of common parameters (**i**–**xii**) associated with NUE and PUE. These are average values of 20 replicates, plotted using GraphPad Prism. Two-way ANOVA was performed to check the significance of the effect of dose, genotypes, and their interaction on the average value of the parameters. *p*-values are summarized with asterisks and calculated with respect to LN (10%N) as control. *p*-values < 0.01–0.05 indicated as *, *p*-values < 0.001–0.01 indicated as **, *p*-values < 0.0001–0.001 indicated as ***, and *p*-values < 0.0001 indicated as **** and ns means non-significant.

**Figure 11 plants-13-02567-f011:**
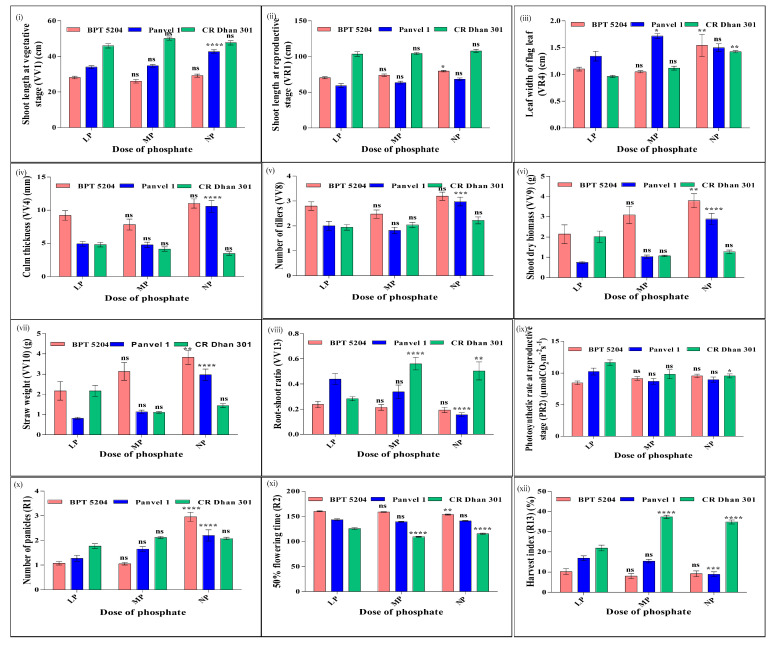
P dose-wise response of common parameters (**i–xii**) associated with NUE and PUE. These are average values of 20 replicates, plotted using GraphPad Prism. Two-way ANOVA was performed to check the significance of the effect of dose, genotypes, and their interaction on the average value of the parameters. *P*-values are summarized with asterisks and calculated with respect to LP (5%P) as control. *p*-values < 0.01–0.05 indicated as *, *p*-values < 0.001–0.01 indicated as **, *p*-values < 0.0001–0.001 indicated as ***, and *p*-values < 0.0001 indicated as **** and ns means non-significant.

**Figure 12 plants-13-02567-f012:**
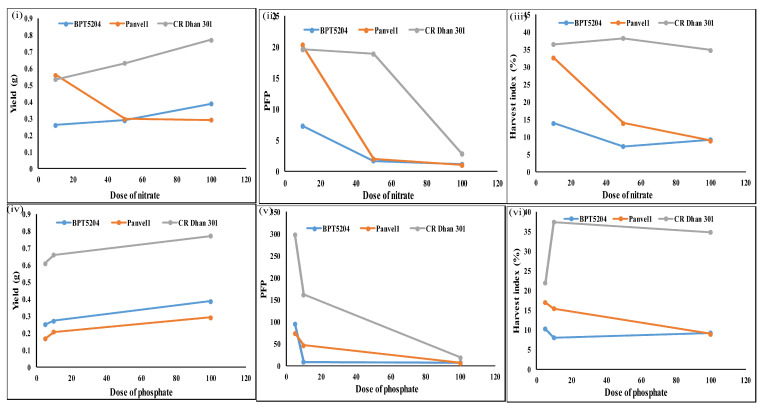
Yield, PFP, and HI of all three genotypes under 3 N and P doses (**i**). Number of filled seeds per panicle under three N doses (**ii**). PFP under three N doses (**iii**). Percentage harvest index under three N doses (**iv**). Number of filled seeds per panicle under three P doses (**v**). PFP under three P doses (**vi**). Percentage harvest index under three P doses. These are average values of 20 replicates, plotted using GraphPad Prism v6.01. Two-way ANOVA was performed to check the significance of the effect of dose, genotypes, and their interaction on the average value of the parameters.

**Figure 13 plants-13-02567-f013:**
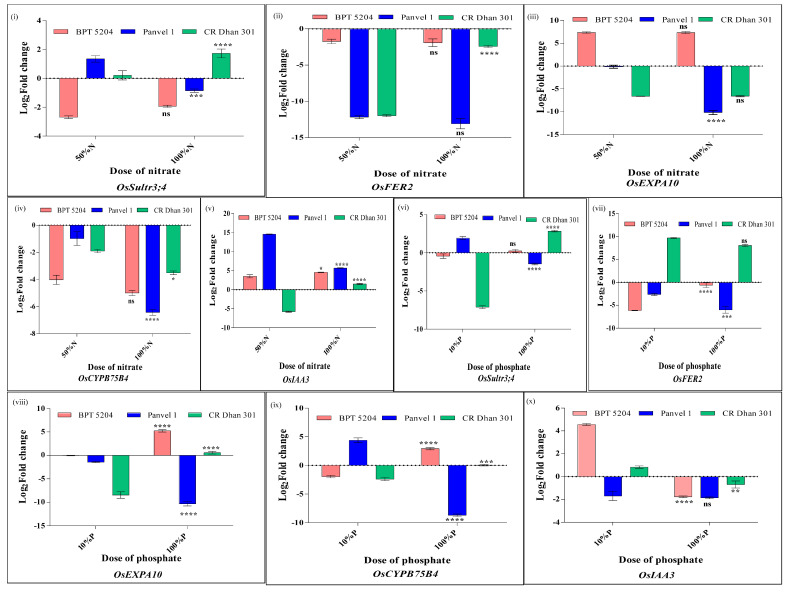
RT-qPCR graphs showing log_2_ fold change values in five NUE/PUE common genes. These are represented as mean ± SE from two biological replicates in all three genotypes grown under (**i**–**v**) 50% N (7.5 mM) and 100% N (15 mM) with 10% N (1.5 mM) as control (**vi**–**x**) 10% P (0.1 mM) and 100% P (1 mM) with 5% P (0.05 mM) as control, *OsFER2* (Os12g0106000), *OsEXPA10* (Os04g0583500), *OsSultr3;4* (Os06g0143700), *OsCYP75B4* (Os10g0317900), and *OsIAA3* (Os01g0231000). Two-way ANOVA was performed to check the significance of the effect of dose, genotypes, and their interaction on the Log_2_ fold change values of the individual genes. *p*-values are summarized with asterisks and calculated with respect to Log_2_ fold change values at medium doses for individual genes. *p*-values < 0.01–0.05 indicated as *, *p*-values < 0.001–0.01 indicated as **, *p*-values < 0.0001–0.001 indicated as ***, and *p*-values < 0.0001 indicated as **** and ns means non-significant.

**Table 1 plants-13-02567-t001:** NUE parameters. Shortlisting of NUE parameters from N-responsive parameters based on yield, partial factor productivity, and harvest index.

N-Responsive Parameters	N-Responsive, Yield Correlated Parameters	N-Responsive PFP Correlated Parameters	N-Responsive Harvest Index Correlated Parameters	NUE Parameters
Shoot length at vegetative stage (VV1)	Shoot length at vegetative stage (VV1)	Shoot length at vegetative stage (VV1)	Shoot length at vegetative stage (VV1)	Shoot length at vegetative stage (VV1)
Shoot length at reproductive stage (VR1)	Shoot length at reproductive stage (VR1)	Shoot length at reproductive stage (VR1)	Shoot length at reproductive stage (VR1)	Shoot length at reproductive stage (VR1)
Chlorophyll content of vegetative stage leaf (VV2)	Chlorophyll content of reproductive stage leaf (VR3)	Chlorophyll content of reproductive stage leaf (VR3)	Chlorophyll content of reproductive stage leaf (VR3)	Chlorophyll content of reproductive stage leaf (VR3)
Chlorophyll content of reproductive stage leaf (VR3)	Leaf width of flag leaf (VR4)	Leaf width of flag leaf (VR4)	Leaf width of flag leaf (VR4)	Leaf width of flag leaf (VR4)
Leaf width of vegetative stage leaf (VV3)	Culm thickness (VV4)	Culm thickness (VV4)	Culm thickness (VV4)	Culm thickness (VV4)
Leaf width of reproductive stage leaf (VR5)	Number of yellow leaves at maturity (VV6)	Number of yellow leaves at maturity (VV6)	Total number of leaves at maturity (VV7)	Number of yellow leaves at maturity (VV6)
Leaf width of flag leaf (VR4)	Total number of leaves at maturity (VV7)	Total number of leaves at maturity (VV7)	Root length (VV12)	Total number of leaves at maturity (VV7)
Culm thickness (VV4)	Root length (VV12)	Root length (VV12)	Root dry biomass (VV11)	Root length (VV12)
Number of green leaves at maturity (VV5)	Root dry biomass (VV11)	Shoot dry biomass (VV9)	Shoot dry biomass (VV9)	Shoot dry biomass (VV9)
Number of yellow leaves at maturity (VV6)	Shoot dry biomass (VV9)	Root-shoot ratio (VV13)	Root-shoot ratio (VV13)	Root-shoot ratio (VV13)
Total number of leaves at maturity (VV7)	Root-shoot ratio (VV13)	Straw weight (VV10)	Straw weight (VV10)	Straw weight (VV10)
Root length (VV12)	Straw weight (VV10)	Number of tillers (VV8)	Number of tillers (VV8)	Number of tillers (VV8)
Root dry biomass (VV11)	Number of tillers (VV8)	Photosynthetic rate of reproductive stage leaf (PR2)	Photosynthetic rate of reproductive stage leaf (PR2)	Photosynthetic rate of reproductive stage leaf (PR2)
Shoot dry biomass (VV9)	Photosynthetic rate of reproductive stage leaf (PR2)	Transpiration rate of flag leaf (PR5)	Transpiration rate of flag leaf (PR5)	Transpiration rate of flag leaf (PR5)
Root-shoot ratio (VV13)	Transpiration rate of flag leaf (PR5)	Transpiration efficiency of flag leaf (PR7)	Transpiration efficiency of flag leaf (PR7)	Transpiration efficiency of flag leaf (PR7)
Straw weight (VV10)	Transpiration efficiency of flag leaf (PR7)	Number of panicles (R1)	Number of panicles (R1)	Number of panicles (R1)
Number of tillers (VV8)	Number of panicles (R1)	50% flowering time (R2)	50% flowering time (R2)	50% flowering time (R2)
Photosynthetic rate of reproductive stage leaf (PR2)	50% flowering time (R2)	Number of unfilled seeds (R5)	Weight of unfilled seeds (R7)	Number of unfilled seeds (R5)
Transpiration rate of flag leaf (PR5)	Number of unfilled seeds (R5)	Weight of unfilled seeds (R7)	Partial factor productivity (R12)	Weight of unfilled seeds (R7)
Transpiration efficiency of flag leaf (PR7)	Weight of unfilled seeds (R7)	Harvest index (R13)		Harvest index (R13)
Number of panicles (R1)	Partial factor productivity (R12)			
50% flowering time (R2)	Harvest index (R13)			
Number of unfilled seeds (R5)				
Weight of unfilled seeds (R7)				
Partial factor productivity (R12)				
Harvest index (R13)				

**Table 2 plants-13-02567-t002:** PUE parameters. Shortlisting of PUE parameters from P-responsive parameters based on yield, partial factor productivity, and harvest index.

P-Responsive Parameters	P-Responsive Yield Related Parameters	P-Responsive PFP Related Parameters	P-Responsive Harvest Index Correlated Parameters	PUE Parameters
Shoot length at vegetative Stage (VV1)	Shoot length at reproductive stage (VR1)	Shoot length at reproductive stage (VR1)	Shoot length at vegetative Stage (VV1)	Shoot length at reproductive stage (VR1)
Shoot length at reproductive stage (VR1)	Leaf width of reproductive stage leaf (VR5)	Leaf width of reproductive stage leaf (VR5)	Shoot length at reproductive stage (VR1)	Leaf width of reproductive stage leaf (VR5)
Chlorophyll content of flag leaf (VR2)	Leaf width of flag leaf (VR4)	Leaf width of flag leaf (VR4)	Chlorophyll content of flag leaf (VR2)	Leaf width of flag leaf (VR4)
Leaf width of vegetative stage leaf (VV3)	Culm thickness (VV4)	Culm thickness (VV4)	Leaf width of flag leaf (VR4)	Culm thickness (VV4)
Leaf width of reproductive stage leaf (VR5)	Root-shoot ratio (VV13)	Root-shoot ratio (VV13)	Culm thickness (VV4)	Root-shoot ratio (VV13)
Leaf width of flag leaf (VR4)	Number of tillers (VV8)	Number of tillers (VV8)	Shoot dry biomass (VV9)	Number of tillers (VV8)
Culm thickness (VV4)	Photosynthetic rate of vegetative stage leaf (PV1)	Photosynthetic rate of vegetative stage leaf (PV1)	Root-shoot ratio (VV13)	Photosynthetic rate of vegetative stage leaf (PV1)
Number of green leaves at maturity (VV5)	Photosynthetic rate of reproductive stage leaf (PR2)	Photosynthetic rate of reproductive stage leaf (PR2)	Straw weight (VV10)	Photosynthetic rate of reproductive stage leaf (PR2)
Number of yellow leaves at maturity (VV6)	Stomatal conductance of vegetative stage leaf (PV2)	Stomatal conductance of vegetative stage leaf (PV2)	Number of tillers (VV8)	Stomatal conductance of vegetative stage leaf (PV2)
Total number of leaves at maturity (VV7)	Transpiration rate of vegetative stage leaf (PV3)	Stomatal conductance of reproductive stage leaf (PR4)	Photosynthetic rate of vegetative stage leaf (PV1)	Transpiration rate of vegetative stage leaf (PV3)
Root length (VV12)	Internal water use efficiency of reproductive stage leaf (PR10)	Transpiration rate of vegetative stage leaf (PV3)	Photosynthetic rate of reproductive stage leaf (PR2)	Number of panicles (R1)
Root dry biomass (VV11)	Number of panicles (R1)	Number of panicles (R1)	Stomatal conductance of vegetative stage leaf (PV2)	Weight of filled seeds (R6)
Shoot dry biomass (VV9)	Number of filled seeds (R4)	Weight of filled seeds (R6)	Stomatal conductance of flag leaf (PR3)	Weight of spikelets
Root-shoot ratio (VV13)	Weight of filled seeds (R6)	Weight of spikelets (R9)	Stomatal conductance of reproductive stage leaf (PR4)	Total panicle weight (R10)
Straw weight (VV10)	Weight of spikelets (R9)	Total panicle weight (R10)	Transpiration rate of vegetative stage leaf (PV3)	Harvest index (R13)
Number of tillers (VV8)	Total panicle weight (R10)	Harvest index (R13)	Number of panicles (R1)	
Photosynthetic rate of vegetative stage leaf (PV1)	Partial factor productivity (R12)		50% flowering time (R2)	
Photosynthetic rate of reproductive stage leaf (PR2)	Harvest index (R13)		Weight of filled seeds (R6)	
Stomatal conductance of vegetative stage leaf (PV2)			Weight of spikelets (R9)	
Stomatal conductance of flag leaf (PR3)			Total panicle weight (R10)	
Stomatal conductance of reproductive stage leaf (PR4)			Partial factor productivity (R12)	
Transpiration rate of vegetative stage leaf (PV3)			Harvest index (R13)	
Internal water use efficiency of reproductive stage leaf (PR10)				
Number of panicles (R1)				
50% flowering time (R2)				
Number of filled seeds (R4)				
Weight of filled seeds (R6)				
Weight of spikelets (R9)				
Total panicle weight (R10)				
Partial factor productivity (R12)				
Harvest index (R13)				

## Data Availability

Data are contained within the article and Appendix A.

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
