# Peer review of "Phenotypic, Physiological, and Gene Expression Analysis for Nitrogen and Phosphorus Use Efficienies in Three Popular Genotypes of Rice (Oryza sativa Indica)"

_plants, 2024, doi:10.3390/plants13182567_

Round 1

Reviewer 1 Report

Comments and Suggestions for Authors

 This manuscript entitled as ‘Phenotypic, physiological and gene expression analysis for ni- 2 trogen and phosphorus use efficiency in three popular geno- 3 types of rice (Oryza sativa indica)’ offers nice info about NUE and PUE in the same studied germplasm. However, I have the following general comments:

1.      Citation needs to be checked to be aligned with the J formatting.

2.      Would recommend give more highlights about the obtained results in the abstract

3.      The introduction must be supported by more hypothesis on the role of both N & P in crop productivity and their negative surpluses use on the environment

4.      Make a list for the 3-rice germplasm used in this study with more pedigree info

5.      In the materials and methods please cite suitable references for the parameters measured.

6.      The results in the way figures are presented are hard to understand and to read even. I suggest authors represent them in a better way, so readers do not suffer to review/read them.

7.      The discussion should be in a sequence based on the way the results are presented

Comments on the Quality of English Language

English with emphasis on the scientific writing needs to be given more attention.

Author Response

Thank you for your constructive comments and suggestions. Accordingly, we have carefully revised the manuscript. Please see the attachment for the responses.

Reviewer 2 Report

Comments and Suggestions for Authors   The manuscript presents a detailed study on the nitrogen (N) and phosphorus (P) use efficiencies (NUE/PUE) in rice, aiming to address the lack of integrated approaches to minimize nutrient wastage and pollution. The authors have conducted a morphophysiological analysis of N/P response in three popular indica genotypes, analyzing a total of 46 parameters across various traits. The study's comprehensive approach to segregate significantly N/P-responsive parameters and correlate them with NUE/PUE indices is commendable. The identification of common high-ranking parameters and the validation of differential N/P responsive gene expression in the three rice genotypes provide valuable insights into the genetic basis of NUE and PUE. However, there are several areas where the manuscript could be improved: 1 The study has clear implications for rice breeding programs aiming to improve NUE and PUE. A dedicated section discussing the practical applications of these findings, including potential strategies for marker-assisted selection or genetic engineering, would be a valuable addition. 2 Suggest incorporating new content into the background section, including cloning of specific genes and their applications. 3 Ensure proper formatting of references. 4 Materials and Methods:Provide further rationale for selecting these three varieties in the Materials and Methods section. Comments on the Quality of English Language

Minor editing of English language required

Author Response

(The authors gave the same response as above.)
